Wikidata and the bibliography of life

Page Roderic D. M. Roderic.Page@glasgow.ac.uk
Institute of Biodiversity, Animal Health and Comparative Medicine, College of Medical, Veterinary & Life Sciences, University of Glasgow , Glasgow , United Kingdom
Casazza Gabriele
Electronic publication date: 2022 Jul 7
Publication date: 2022
Volume: 10
Electronic Location ID: e13712
Received 2021 May 17; Accepted 2022 Jun 20
Copyright: © 2022 Page
Copyright year: 2022
Copyright holder: Page
License: This is an open access article distributed under the terms of the Creative Commons Attribution License, which permits unrestricted use, distribution, reproduction and adaptation in any medium and for any purpose provided that it is properly attributed. For attribution, the original author(s), title, publication source (PeerJ) and either DOI or URL of the article must be cited.
License URL: https://creativecommons.org/licenses/by/4.0/

Keywords: Wikidata, Taxonomy, Knowledge graph, Bibliometrics, Crowd sourcing

Funding: The author received no funding for this work.

==============================
Biological taxonomy rests on a long tail of publications spanning nearly three centuries. Not only is this literature vital to resolving disputes about taxonomy and nomenclature, for many species it represents a key source—indeed sometimes the only source—of information about that species. Unlike other disciplines such as biomedicine, the taxonomic community lacks a centralised, curated literature database (the “bibliography of life”). This article argues that Wikidata can be that database as it has flexible and sophisticated models of bibliographic information, and an active community of people and programs (“bots”) adding, editing, and curating that information.

Introduction

Much of the primary data about the planet’s biodiversity is contained in the taxonomic literature, a corpus that dates from the eighteenth century. Whereas other biological disciplines have created substantial bibliographic databases, such as PubMed, and open access repositories for work sponsored by specific funding agencies and charities agencies, such as Europe PubMed Central (The Europe PMC Consortium, 2015), the taxonomic literature mostly lingers in relative obscurity (Page, 2016a). There are several projects trying to redress this problem by digitising the taxonomic literature, ranging from global initiatives such as the Biodiversity Heritage Library (BHL) (Gwinn & Rinaldo, 2009) to extensive, regional repositories such as the Zoological-Botanical Database (ZOBODAT) (Gusenleitner & Malicky, 2017). While the bulk of BHL content comprises legacy works that are out of copyright, recently this has been supplemented by an influx of more recent content so that BHL is no longer “legacy only”. A complementary initiative, the Biodiversity Literature Repository (BLR) is focussed on recently published born digital content and its component parts, such as figures and taxonomic treatments (Egloff et al., 2017). Taxonomy also benefits from digitising initiatives that do not specifically target the taxonomic literature but which include taxonomic journals, such as E-Periodica (Wanger & Ehrismann, 2016).

Digitisation greatly increases the accessibility, but not necessarily the discoverability of content. The Biodiversity Heritage Library has scanned volumes for many journals, but unless articles contained within those volumes are indexed those articles will be difficult to find. This was the motivation for my BioStor project (Page, 2011), which to date has extracted over 200,000 articles from content scanned by BHL. Another impediment to discoverability is the widespread taxonomic practice of using “micro-citations”, that is, citing a page or set of pages within a work, rather than the work itself (Page, 2009). Experts in a particular group are usually familiar with these micro-citations, but non-experts may find them challenging to interpret.

Discoverability of the taxonomic literature would be greatly improved if we had a single, easily accessible database of all taxonomic publications (King et al., 2011). While taxonomy has some highly visible journals, there is a long tail of taxonomic publication in small, often obscure journals (Page, 2016c). Not only does lack of discoverability hamper taxonomic research, it also hampers recognition of the value of that research. Taxonomists have long complained that standard measures of academic impact do not work well for taxonomists (Garfield, 2001), and the ranking of major taxonomic journals by commercial organisations such as Clarivate can undergo dramatic and seemingly capricious changes (Hamilton et al., 2021). A commonly proposed remedy is increased citation of taxonomic work (Werner, 2006), such as original descriptions of new species. Regardless of the merits of these proposals, they founder when confronted with the practical issue that we don’t have citable references for many, if not most, species descriptions.

The challenge of discoverability is not unique to taxonomic literature. There have been long standing calls for what Cameron (1997) described as a “universal citation database”. Recent developments such as the OpenCitations infrastructure (Peroni & Shotton, 2020) and the WikiCite project have brought us considerably closer to this goal. Indeed, in the last few years there has been a growing effort to add bibliographic details for the entire academic corpus to Wikidata, an open database of structured information (Waagmeester et al., 2020). Bibliographic metadata is at the heart of measures of academic performance and impact, and these measures are typically provided from closed data held by commercial organisations (Aspesi & Brand, 2020). Having an open bibliographic database for taxonomy leads to the possibility of more transparent analytics for the discipline.

In this article I make the case for Wikidata as the logical venue for a global database of taxonomic literature, the so-called “bibliography of life” (King et al., 2011). Given that this may exclude much of the literature on medicine, agriculture, genomics, etc. this may seem an overly narrow definition of what constitutes a bibliography of life. But I argue that the term is justified given the taxonomic breadth of such a bibliography. The effort devoted to studying different taxa is very uneven, such that in species-rich groups such as Coleoptera (beetles) an individual species may be the subject of a publication only once every 100 years (May, 1988). This uneven coverage is only likely to increase with the growing importance of citizen science (Troudet et al., 2017) and the increasing dominance of research on model organisms (Farris, 2020). For many species the taxonomic literature will be the best (possibly the only) source of published information on that species, hence arguably the only database that could claim to be a bibliography of life is one that includes all taxa, that is, it includes all the taxonomic literature.

In this article, I make the case for constructing the bibliography of life using Wikidata. I begin by providing background on Wikidata, describing how it models bibliographic data, and how it can be populated with data. I then summarise some analyses that assess the extent to which the Wikidata community curates bibliographic data, and estimate the “density” of the Wikidata knowledge graph for bibliographic data

Wikidata

Wikidata is a store of structured information or “statements” about things or concepts (“items”). Each statement comprises a key-value pair where the key is a community-defined property, and the value is editable by any Wikidata user. Each Wikidata item has a unique identifier of the form Qn (where n is an integer), each property has an identifier in the form Pn (in this article I often refer to Wikidata properties by their P number). A given key-value pair can have one or more qualifiers (Vrandečić & Krötzsch, 2014), that is, a statement about that particular value. For example, a multi-author publication will have multiple values of the property “author” (P50). Adding the qualifier “series ordinal” (P1545) to each value enables us to express order of authorship, i.e., the first author has a series ordinal qualifier of “1”, the second author has the value “2”, and so on.

Ideally values in Wikidata are accompanied by one or more references to the sources of those values. Typically references are links to external sources (such as a web site or database), but they can also be links to another item in Wikidata (for example the item corresponding to a publication that is the source of that value). Among the strengths of Wikidata is its support for multiple languages, and for multiple values for the same property. Hence Wikidata can accommodate cases where there is legitimate disagreement about the value a property should take (for example, the date of publication of a work). While any user can edit values, properties are added by community consensus. A property is proposed, discussed, and if it receives community support it becomes available for editors to add to an item. The information stored in Wikidata can be expressed as Resource Description Framework (RDF) triples (Erxleben et al., 2014) and there is a SPARQL (SPARQL Protocol and RDF Query Language) endpoint that enables anyone to query the data.

Wikicite

The original scope of Wikidata was to provide structured data to underpin the different Wikipedia projects. Hence, notionally each item in Wikidata had a corresponding entity in at least one of the various Wikipedias. However, as Wikidata has grown the potential of having a single, queryable, community-edited database of structured information has become increasingly clear. Hence many items being added to Wikidata might not themselves have a Wikipedia page but are relevant to the content and goals of Wikipedia. A good example of this are bibliographic citations, which are a key source of support for factual statements made on Wikipedia.

The Wikicite project started out with the goal to provide structured bibliographic data for citations across the different Wikipedia projects. Given that the scope of Wikipedia includes taxonomy, many of the publications cited in Wikipedia (and hence destined to be in Wikidata) are relevant to taxonomy. Furthermore, there is a wiki devoted entirely to taxonomy (Wikispecies), which includes pages for taxa, taxonomists, and taxonomic publications. Many of these pages also have corresponding items in Wikidata. Hence a considerable amount of taxonomic literature has already been added by contributors to the WikiCite project.

Data contributions to Wikidata typically come in two forms, manual edits by individual people or automated edits by software (“bots”). A number of bots add bibliographic metadata sourced from databases such as PubMed and CrossRef. For example, given a CrossRef DOI for an article the CrossRef API can be used to retrieve the metadata for the corresponding article. If one wanted to include only publications cited by Wikipedia, one would then need a list of DOIs cited on Wikipedia pages. Alternatively, one could proactively add articles with DOIs to Wikidata even if they aren’t currently cited on Wikipedia, on the assumption that as Wikipedia grows it is likely that more and more articles will be cited. This means it is a short step to expanding the scope to include most, if not all of the academic corpus in Wikidata. One motivation for this is to have openly accessible bibliographic data which can be used to enable freely accessible measures of the activity and impact of researchers (Nielsen, Mietchen & Willighagen, 2017).

As a consequence of work done by the WikiCite community, and the prominence of taxonomy in Wikipedia and Wikispecies, Wikidata already contains a considerable number of publications relevant to taxonomy. This, coupled with the sophistication of the data model, powerful query language, and the existence of an enthusiastic community of editors makes a strong case for Wikidata being a promising platform for a bibliography of life.

Bibliographic data in Wikidata

The Wikidata model for a publication has evolved over time as the community adds properties and recommendations for their use. Figure 1 shows how a scientific article can be modelled in Wikidata.

Figure 1 Simplified representation of a scholarly article in Wikidata.

The article (Bruyns, Mapaya & Hedderson, 2006) corresponds to Wikidata item Q28960244. Statements about this item are made using Wikidata properties (indicated by edges in the graph labelled with the prefix “P” followed by a number). Statement values that are simple strings (e.g., title, volume, paging, and DOI) are enclosed in yellow boxes. Some statements connect Wikidata items together, such as that labelled P1433 which connects the item for the article (Q28960244) to the item for the journal in which it was published (Q2003024). Some statements have qualifiers that provide details about that statement (for example, its order in a list), these statements are represented by empty circles linked to the statement value and its qualifiers.

Wikidata items are given one or more types using Wikidata property P31 (instance), such as Q13442814 for a scholarly article, and Q571 for a book. There are properties for the typical metadata associated with an article, such as title, journal that contains the article, volume, pagination, and date of publication. Wikidata supports values in multiple languages, so that articles with titles in multiple languages can have all those titles represented. Authorship is handled in two distinct but complementary ways. If an author of a publication is known to have a Wikidata entry then the author property (P50) links the item for the publication to the item for that author. If it is not known whether the author exists in Wikidata their name can be stored as a simple string value (P2093). In Fig. 1 there are examples of both authors. There are tools available to subsequently map those name strings to the corresponding Wikidata items.

External identifiers, such as ones provided by the publishing industry (e.g., DOIs), archiving services (e.g., Handles), and domain-specific databases (such as PubMed, ZooBank, etc.) can also be added to the Wikidata item. Wikidata items are being decorated with an increasing number of diverse identifiers, hence Wikidata is increasingly playing a role as an “identity broker” enabling cross-links between identifiers from different databases (Van Veen, 2019).

Links between publications

Publications rarely exist in isolation from each other, hence we can connect them using a range of properties. The most obvious relationship is citation, where one publication cites another. Adding this information helps flesh out the citation graph, enables us to track the provenance of an idea, and also discover potentially related publications through co-citation (Marshakova-Shaikevich, 1973; Small, 1973).

Other relationships supported by Wikidata include errata where one publication corrects errors or mistakes in a previous publication, and translations, where a publication may exist in more than one language. For example, the article Korotyaev (2018) is an English translation of Коротяев (2018), the corresponding items in Wikidata can be connected by properties reflecting that relationship.

Links to facts

A key motivation for including publications in Wikidata is to provide trustworthy sources of references for statements made in Wikidata. For example, statements about the birth and death dates for a person, the exact date of publication of a work, the date at which a journal changed its name, or the publication of a taxonomic name can all be supported by adding references to the relevant source.

As an example, the taxonomic name Euphorbia bicompacta Bruyns was published in Bruyns, Mapaya & Hedderson (2006) as a replacement for the name Synadenium compactum N.E.Br. This publication (Q28960244) is the one discussed above in Fig. 1. The Wikidata item for Euphorbia bicompacta (Q5851419) has a property “taxon name” (P225) with the value “Euphorbia bicompacta” and Wikidata item Q28960244 as a reference for that value (see Fig. 2).

Figure 2 Example of a publication being used as a reference to support a statement in Wikidata.

The taxon Q5851419 has the taxon name (P225) “Euphorbia bicompacta”. The reference for this value comprises the Wikidata identifiers for a publication (Q28960244), the location in that publication where the name occurs (page 412), and the role the publication plays (publishing a replacement name).

Populating Wikidata

Creating a bibliography of life would only be conceivable if much of the work of populating it could be automated, and if freely accessible sources of data were available. Bibliographic metadata from CrossRef and PubMed are constantly being added by automated tools (“bots”). This means that many publications that have a CrossRef Digital Object Identifier (DOI) or have an entry in PubMed are likely to be already in Wikidata. If they aren’t, then it is straightforward to add them. Data from these sources are typically of high quality, although sometimes the data is limited or incorrect, for example, in not including lists of literature cited, or there may be typographic or character encoding errors in the data. An advantage of a community-editable resource is that these can be found and subsequently corrected by the community.

While much of the biomedical literature, and an increasing fraction of modern taxonomic literature has CrossRef DOIs, much of the taxonomic corpus either lacks a DOI, or may have a DOI issued by a registration agency other than CrossRef. The DOI foundation has several members that issue DOIs, and these differ in the support they provide for resolving DOIs to machine readable data. CrossRef DOIs can return extensive metadata about an article in CiteProc JSON, a default standard for bibliographic metadata (Willighagen, 2019; Bennett, 2021). Some DOI agencies support CiteProc (albeit not as fully populated as CrossRef), however agencies such as ITISC—which is issuing DOIs for many Chinese articles (Wang et al., 2018)—do not support machine readability at all. Hence, not all DOIs are equally easy to work with.

There are also publications with persistent identifiers that are not DOIs (such as Handles), publications which lack persistent identifiers but are online, and publications which may not be online at all. There are various strategies we can use to gather bibliographic data for these publications. Below I describe some of these strategies. Source code for some of these approaches is available at https://github.com/rdmpage/wikidata-bibliographic-data.

Scrape metadata from the web

Web sites for some journals contain embedded machine-readable metadata about publications in their web pages to enhance discoverability by search engines such as Google Scholar. These tags also enable software tools (e.g., reference managers such as Zotero) to easily extract bibliographic metadata to be stored by users of those tools. Although typically there are journal and publisher-specific idiosyncrasies in how the metadata is marked up, it is relatively straightforward to write software to fetch these web pages and extract the metadata.

Lists of literature cited

Most articles will have a list of literature cited, so when taxonomists publish their work they are also continually publishing bibliographic metadata. These lists are becoming increasingly accessible to machines. Furthermore, CrossRef is encouraging publishers to include lists of references cited in their submissions to CrossRef. If both the citing article and the cited article have CrossRef DOIs and the citing article references are submitted to CrossRef, then this citation link will appear in the next update of COCI, the OpenCitations index of open Crossref DOI-to-DOI citations (Heibi, Peroni & Shotton, 2019). While this helps grow the citation network, it overlooks all those publications that lack DOIs (or which lacked them at the time the citing article was published). However, the metadata for references cited which lack DOIs can still be used to help populate Wikidata.

Some publishers provide article text in machine-readable formats such as XML where the references are identified and can be easily extracted. Other publishers may provide lists of references in the web view of an article, sometimes with embedded markup. Hence, we can regard taxonomists as, in effect, “crowd sourcing” the taxonomic literature simply by the act of publishing their research. For example, articles published in the journal Zootaxa together contain over a million references cited (Page, 2020).

Taxonomic databases

The numerous taxonomic databases being developed by the community, often focussed on a particular taxonomic group, are yet another source of bibliographic data. Regrettably, in many cases taxonomic databases do not treat the taxonomic literature as a first-class citizen, and hence the data may be stored in an abbreviated form (such as the micro-citations mentioned above). But some databases do provide high-quality curated literature which can be used to help populate Wikidata.

Databases of researchers

Yet another potential source of data are the collections of articles created by researchers as part of an online profile or identity, such as ORCID (https://orcid.org/) or ResearchGate (https://www.researchgate.net/). Using a combination of manual input and web services, ORCID assembles a list of publications (and other outputs) linked to a researcher’s unique identifier (their ORCID id). This data is openly available via an API. In contrast, ResearchGate is a commercial website where members can upload lists of their publications, and provide access to the publications themselves (on the understanding that their members have the legal right to do so). Although ResearchGate is “closed” in that it lacks a publicly available API, they do embed structured markup in their web pages which links authors to their publications using terms from the https://schema.org/ vocabulary.

Wikis

The sources which perhaps most closely match the notion of “crowd sourcing” are Wikidata itself, and other wikis of the Wikipedia Foundation, such as Wikipedia and Wikispecies. Indeed, in much the same way that we can regard Wikipedia as an Encyclopaedia of Life (Page, 2010), Wikispecies can be regarded as a crowd sourced “bibliography of life” where volunteers are assembling a wiki with one page per taxon, often including extensive lists of references cited. However, these references are often entered as simple text strings with little or no structured markup, making it challenging to extract structured metadata, and hence limiting the utility of Wikispecies.

Full-text

Wikidata stores metadata rather than full-text content, that is, it stores information about a publication, not the contents of the publication itself. A growing proportion of the taxonomic literature is being digitised, such that articles may be available in formats such as PDF or sets of images (e.g., scans of printed works). Given the alarming ease with which links to online content can break (Laakso, Matthias & Jahn, 2020) a convention on Wikidata is to include not only a link to a freely available PDF but also a link to an archived version, e.g. on the Internet’s Wayback Machine (https://web.archive.org/). Another strategy (one that I have regularly used) is to store a copy of the PDF on Internet Archive itself and include the Internet Archive identifier as a property of the publication on Wikidata.

Other ways to access content include tools that take a DOI and return a PDF if one is available online, either freely available, e.g. Unpaywall (http://unpaywall.org/) or “pirated” (Bohannon, 2016). Some publishers such as the China National Knowledge Infrastructure (CNKI) have mobile phone apps that provide access to their content through that app.

Being able to access the content of the articles themselves not only means that we can read the article, but it also provides a way to augment existing metadata. In my own experience key data such as page numbers were often not recorded in the available metadata for an article. This can make it harder to link publications to taxonomic names using “microcitations”, where the only information we have is a journal, a volume, and a page number. However, if we have access to a digital version of the article we can extract the page numbers. This need not be a manual process, for instance the Internet Archive generates a file for each PDF that contains a best-guess of the page numbers in the PDF. We can use those to add missing pagination values to the corresponding Wikidata items.

Exploring bibliographic data in wikidata

A key goal for the bibliography of life is to be able to link every taxonomic name for eukaryote species to its original description using a unique identifier (e.g., a DOI) and ideally a link to a digitised version of that publication. The scale of this challenge was discussed in (Page, 2016a), and an attempt to do this for animal names led to my BioNames project (Page, 2013). I have done similar work for plants and fungi based on the International Plant Name Index (IPNI) (https://ipni.org/) and Index Fungorum (http://www.indexfungorum.org/), a subset of which has been released on GBIF (Page, 2016b), and published as a both a “datasette” (Page, 2018) and raw data dumps (Page, 2020). Based on this work across animals, plants, and fungi, a little under four million taxonomic names have associated bibliographic metadata (Table 1), such as a citation to a publication or a page in a publication. Depending on taxonomic group, anywhere between 20–40% of those citations have been mapped to an external identifier such as a DOI, and some 16–25% of taxonomic names have their associated publication in Wikidata. The 880,000 links between names and Wikidata publication items correspond to just under 200,000 distinct publications. A random sample of 10,000 of these publications was used in the analyses described below.

Table 1 Current progress towards mapping taxonomic names to the source literature. For each database the table gives the number of names that have an associated bibliographic reference, the number of those that have at least one external identifier, and the number of those that are known to be in Wikidata.

Database	Number of taxonomic names with publications	Number of names with publication identifier	Percent with identifier	Number of names with publications in Wikidata	Percent in Wikidata	
BioNames	1,715,602	747,983	43.6	432,370	25.2	
IPNI	1,708,187	512,985	30.0	378,616	22.2	
Index Fungorum	444,235	93,871	21.1	72,824	16.4	

A community of editors

One of the challenges in community-based editing of scientific data is assembling that community. We could create a domain-specific database and hope a community coalesces around that database. Alternatively, we take the data to where an active community already exists. This is the approach taken by projects such as Gene Wiki (Good et al., 2012). If Wikidata is going to be the place to assemble the bibliography of life, a natural question is: “Does the community actually edit taxonomic publications?” To assess this, I looked at the edit history of the random sample of 10,000 Wikidata items generated above. For each of these items I retrieved the number of edits made since the Wikidata item was created, when those edits were made, and what properties were edited.

Figure 3 visualises the edit history for the sample of 10,000 publications as a scatter plot of creation timestamp against edit timestamp. If an item was only edited at the time it was created then all points would fall along the diagonal and the lower right triangle in Fig. 3 would be empty. This diagonal continues to go up and to the right as time goes on. Any edit to an item appears as a dot to the right of the dot on the diagonal that represent the item’s creation. If there are no dots to the right of the diagonal, then an item has not been edited since its creation. Figure 3 shows that many items undergo a series of sporadic edits over time. Some of these edits occur shortly after item creation. For example, there are Wikidata bots whose function is to add a description for a new item in a specific language. Other edits may happen later in the life cycle of an item, for example if a user associates a publication with its author or links a publication to its main subject. Or there may be a bulk update of many items by a bot that edits a specific property.

Figure 3 Edit history for 10,000 Wikidata publications.

The x and y axes are in units of Unix timestamps (seconds since 1 January 1970), each point in the chart is an individual edit of an item, where the value of x is the timestamp of that edit, and the value of y is the timestamp for when that item was first created.

The most common edits observed in the publications involved the authors of those publications (properties P2093 and P50), as well as adding values for P921 “main subject” (a form of tagging an item) (Fig. 4). Edits in Wikidata can be made by people, either directly by editing a record in Wikidata, or using bulk tools such as Quickstatements (https://quickstatements.toolforge.org/#/). Edits can also be made by automated programs (“bots”). Of the top ten editors of publications, half are bots (Fig. 5).

Figure 4 Frequency of Wikidata property edits.

Number of edits made for each property in the sample of bibliographic items shown in Fig. 3.

Figure 5 Most active editors of publications on Wikidata.

Plot of the number of edits of bibliographic items made by a given user for a sample of 10,000 bibliographic items. Users are colour-coded by whether they are bots or people.

This approach to measuring edit activity assumes that only edits made to an item itself are relevant. However edits may be made to other items that link those items to the current item. For example, adding a “cites work” statement to an item does not result in any changes to the item being cited (i.e., the target of the “cites work” statement).

Knowledge graph density

Conceptually a knowledge graph comprises entities (nodes) that are connected by facts (edges). The number of facts for an entity is a measure of the knowledge graph’s density, which for many graphs is low, often averaging less than two facts per entity (Hegde & Talukdar, 2015). Note that this definition of “facts” ignores simple statements associated with an entity (e.g., the number of pages in an article). These are also facts in the sense of being statements about an entity, but we don’t need a knowledge graph to store them. The true power of a knowledge graph comes from the density of the connections between entities.

To assess the connection density of bibliographic entities in Wikidata, I counted the number of links between bibliographic items and other Wikidata entities in the sample of 10,000 bibliographic items. In counting these connections some entities, such as those for language, were not counted to avoid inflating the estimate of knowledge density based on what are essentially administrative metadata. The properties that were counted are shown in Table 2.

Table 2 Frequency of linking properties between Wikidata items. Counts of the different Wikidata properties that linked two or from the sample of 10,000 taxonomic publications.

Property	Frequency	
Published in (P1433)	9,667	
Cites work (P2860)	30,357	
Author (P50)	3,699	
Stated in (P248)	4,655	
Sponsor (P859)	93	
Corrigendum/erratum (P2507)	13	
Part of (P361)	7	
Place of publication (P291)	10	
Has part or parts (P527)	1	

The average link density for the sample of publications was 4.17, with the modal number of connections being one. Hence this part of the knowledge graph is relatively sparse, with most publications having just the connection to a parent publication (typically a journal). Some publication items are connected to other items via citation relationships, either as the source or the target of that relationship (i.e., citing or cited by).

Author coverage

The bulk of publications added to Wikidata treat authors as “strings” not “things”, that is, most authors are listed as names using the P2093 “authors name string” property, rather than as Wikidata items using the P50 “author” property (see Fig. 1). Ideally all authors of publications would be Wikidata items, not simply text strings, and indeed making that conversion is among the most commonly made edits (Fig. 4). Realising this goal requires that all authors of taxonomic publications have items in Wikidata, which in turn is part of a broader goal of having a Wikidata item for everyone involved in taxonomic research (Groom et al., 2020).

There are several databases of taxonomists that have representation in Wikidata, although their coverage in Wikidata is variable. For example, the International Plant Names Index (IPNI) contained approximately 43,000 authors in 2013 (Lindon et al., 2015), and currently some 53,000 Wikidata items have IPNI author IDs. At the time of writing (2021) ZooBank (Pyle & Michel, 2008) contains some 87,000 authors, of which 17,000 are in Wikidata. The Biodiversity Heritage Library has 28,500 authors in Wikidata, while Wikispecies contributes 61,000 authors to Wikidata. There is overlap among these sources. For example, almost all of the ZooBank authors that are in Wikidata are also in Wikispecies, whereas the majority of authors sourced from IPNI are unique to IPNI (Fig. 6). What is unclear is how much of the lack of overlap between authors in the different sources databases is real (do they represent different sets of authors?), vs. a lack of mapping between identifiers (how many records are for the same people, just using different identifiers?). There is considerable scope for reconciling authors between these databases, as well as other sources of information on people, such as ORCID and ResearchGate. It is not enough to merely have authors represented in Wikidata, we also need to link them to their publications. The source databases (BHL, IPNI, Wikispecies, and ZooBank) all contain links between authors and their publications, and much more use could be made of these sources to add P50 author links (Page, 2019).

Figure 6 Overlap in authors from different sources.

Overlap among four different data sources for authors in Wikidata (BHL, IPNI, Wikispecies, and ZooBank). Each circle is labelled by the corresponding source database and the total number of authors that source contributes to Wikidata. Authors may have entries in multiple databases, these are represented by the overlapping regions in the Venn diagram.

Discussion

By providing a robust, open platform for community editing of structured data, Wikidata seems an ideal platform for the bibliography of life. It not only benefits from a community of active editors, it piggy backs on the remarkable fact that taxonomy is the only discipline to have its own Wikimedia Foundation project (Wikispecies). Consequently, a large number of taxonomic works and their authors already exist in Wikidata. As more and more taxonomic publications acquire DOIs, and as more working taxonomists acquire ORCID ids, the taxonomic literature component of Wikidata will automatically grow as content linked to these identifiers is routinely harvested by Wikidata bots. This leaves a large fraction of the taxonomic literature to be added by other means, but as discussed there are numerous ways to do that. It is not unreasonable to expect that the bulk of the taxonomic literature will find its way into Wikidata in the next few years.

Wikidata has a higher density that most knowledge graphs (Hegde & Talukdar, 2015), highlighting the importance of having an active community of editors. However, being a community project, Wikidata has a number of quirks. It is possible for people working independently to create multiple Wikidata items for the same thing (although there is a simple mechanism for merging such duplicates). The way Wikidata models a given class of entities (such as “taxa” or “books”) is determined on an ad hoc basis by a self-assembling community of interested people. This can lead to multiple ways to do the same thing, which presents challenges to both editing and querying the data. While these quirks would be less likely in a domain-specific database, it is unlikely that such a database would have the level of community engagement we see in Wikidata.

Given that the bibliography of life is of little use unless it has content, I have focussed here on where that content comes from, and to what extent the Wikidata community contributes to the curation and improvement of that content. There is considerable scope for analysing gaps and biases in coverage in geography and language (Miquel-Ribé & Laniado, 2021) as well as taxonomy (Grieneisen et al., 2014; Troudet et al., 2017). Wikidata’s user interface is aimed at data entry and editing rather than search and visualisation. Creating engaging, user-friendly interfaces (Whitelaw, 2015) to navigate the bibliography of life is major challenge which will be addressed elsewhere.

Supplemental Information

Supplemental Information 1 List of Wikidata items for sample of taxonomic publications.

Click here for additional data file.

Supplemental Information 2 Edit timestamps for sample of taxonomic publications.

Click here for additional data file.

Supplemental Information 3 Edit counts for Wikidata users.

Click here for additional data file.

Supplemental Information 4 Edit counts by property.

Click here for additional data file.

I thank the numerous Wikidata contributors (some known only by their usernames) who have helped me learn the ropes and navigate the active and opinionated Wikidata community. Among these contributors are Christian Ferrer, Siobhan Leachman, Andy Mabbett, Daniel Mietchen, Succu, Andra Waagmeester, and Egon Willighagen. Lars Willighagen provided helpful feedback on an earlier draft. I am grateful to David Shotton, John Mittermeier, and an anonymous reviewer for their helpful critiques of the manuscript.

Additional Information and Declarations

Competing Interests

Author Contributions

Data Availability

The author declares that they have no competing interests.

Roderic D. M. Page conceived and designed the experiments, performed the experiments, analyzed the data, prepared figures and/or tables, authored or reviewed drafts of the article, and approved the final draft.

The following information was supplied regarding data availability:

The raw data and code are available at GitHub: https://github.com/rdmpage/alec.

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
