# Peer review of "Wikidata and the bibliography of life"

_PeerJ, doi:10.7717/peerj.13712_

## Round 0.1 · original submission · Major Revisions

Dear Dr. Page,

According to the reviewers, the manuscript provides an interesting examination of whether Wikidata is suitable as a global repository and structured database for taxonomic publications. However, they found some major weaknesses.

The first major issue concerns the structure of the manuscript. In particular, a reviewer finds that the manuscript is poorly focused towards a specific agenda/target and suggests adding gaps and limitations of Wikidata and how these could potentially be addressed.

The second methodological issue concerns the sample of 1000 bibliographic entities and the representation of the edit data on these entities presented in Fig. 7.

So, I encourage you to improve the manuscript according to the reviewers' suggestions. Please, respond point-to-point to the comments of reviewers to speed up the process of revision.

Once again, thank you for submitting your manuscript to PeerJ and we look forward to receiving your revision.

Sincerely,
Gabriele Casazza

·

Basic reporting

This short article presents a detailed examination of whether Wikidata is a suitable repository for bibliographic information relating to scholarly papers describing the taxonomy of biological species. The “life” of the article's title thus has a very precise and restricted meaning of biological species and their taxonomic organization, and totally excludes other areas of biology such as animal behaviour, gene function or cellular ultrastructure. The shorthand phrase “the bibliography of life” is clearly defined to have this restricted meaning by the author, and is permissible as being memorable.

The article is clear, well-written, properly referenced and well supported by tables, figure and raw data files, and thus meets all the criteria required by PeerJ in terms of basic reporting. There are a number of minor typographical corrections and suggestions for changes of wording or emphasis that I have noted as Comments within the PDF of the article itself (attached). These, together with more serious points noted below concerning the sample of 1000 bibliographic entities and the representation of the edit data on these entities presented in Fig. 7, and my suggestion for two concluding tables, must be addressed by the author before the revised article is resubmitted.

Experimental design

Unlike most PeerJ papers that report experimental research directly into biological phenomena, the research reported in this article is into the availability of bibliographic information contained (or potentially contained) within Wikidata relating to papers describing biological species. It thus falls somewhere between a Research Article and a Literature Review Article, the two types of report permitted to appear within PeerJ. The conclusions of the paper are highly relevant to biology and 'open science' as a whole, and to modern on-line methods of discovering information about individual species, which makes PeerJ a very suitable journal for its publication. Lest there be any hesitation on the part of PeerJ editors about its suitability for PeerJ, it should be noted that the taxonomic publications that are the subjects of the investigation reported in this article are themselves not “research” in the sense of experimental investigations, but rather are descriptive observational reports about the existence of species. Thus these publications too, which lie at the very heart of biology, lie outside our conventional understanding of “experimental biology”.

Because the article is not a typical experimental research report, this review is also atypical, particularly in this Experimental Design section.

In evaluating this article, it is important to realize that the author is both a specialist in biological taxonomy and an expert user of Wikidata. He is thus ideally placed to write on the subject of the article.

The body of the article falls into two main parts.

The first part, “Wikidata”, is a clear and helpful description of Wikidata, explaining how information is recorded within it and made freely available to scholars and the general public, the original sources of such information (particularly of taxonomic and bibliographic information), how items of such information are entered and may subsequently be edited either by people or 'bots', and the special factors that make this information repository suitable for bibliographic information relating to scholarly papers describing the taxonomy of biological species.

The second part, “Exploring Bibliographic Data in Wikidata”, is an investigation of the extent to which Wikidata lives up to its potential in providing a comprehensive repository for the bibliographic information under review, and includes an honest evaluation of the shortcomings and problems inherent in the Wikidata method of information provision and editing by an unpaid community of experts volunteering their time and expertise.

In this second section, the author first describes open software he himself has developed that provides the functionality within a simple website called ALEC (All Literature Electronically Catalogued), giving the user convenient access to the bibliographic information recorded within Wikidata, and illustrates how ALEC returns different types of data.

He then evaluates the Wikidata information for a sample of 1000 bibliographic entities recorded in Wikidata, in terms of the frequency of edits of these records, their citation links to other publications, etc., before going on to look at the taxonomic and author coverage for Wikidata as a whole. To the extent that this article involves experimental investigation, it is here in this section, and it is here that my main criticisms of the article rest.

First, the author announces his analysis of the sample of 1000 publications without describing how this sample of 1000 was selected from all the relevant publications within Wikidata, and whether the method of selection might influence the results obtained. This is a very serious omission that must be rectified. Ideally, one would like to see the same analyses run on multiple samples of 1000 publications, but in reality I don't think this would add anything significant to the paper, since it is the general trends revealed by the analyses that are important, rather than particular numerical values.

Second, in Figure 7, the figure that shows the number of edits each of this 1000 publication entries in Wikidata has received, the initial entry date for the information about each publication is indicated on the figure diagonal by a dot. Unfortunately, the size of the dot used is so great that the figure appears to show only about 100 publications, not the full 1000, since individual dots overlap. At least, I assume that to be the explanation for the small number of apparent dots. The author should discuss this overlap problem, and should fix it by revising the figure. Either smaller dots with different colours should be used, or an electronically zoomable figure should be provided, to enable the results for different publications to be individually visible.

Third, in the discussion of the data in Fig. 7, there appears to be an implicit assumption that, following initial data entry, a large number of edits on the information relating to each publication is a 'good thing'. However, if the information was fully and correctly entered initially, no subsequent editing of the entry would be required. The author should explicitly address this point.

Validity of the findings

With the exceptions of the problems relating to the sample of 1000 publications and to Figure 7, discussed above, the conclusions drawn from this investigation of Wikidata are clearly presented and straightforwardly discussed.

Additional comments

Within the presentation of the results of his investigation, and in the concluding Discussion, the author has clearly and honestly set out both the advantages and also the present limitations and quirks of Wikidata as a potential repository for the Bibliography of Life. Of these, the most important limitations are the lack of an adequate data model that separates taxa and their names, the possibility for different editors to create duplicate entries for the same publication, the recording of many authors as name strings rather than as entities with names, the current lack of comprehensive coverage of the field, the lack of persistent identifiers (e.g. DOIs, Digital Object Identifiers) for much of the taxonomic literature, and the limited size of the community of taxonomists currently working with data in Wikidata, which limits the rate of progress.

The potential for Wikidata to become in reality the repository to document the Bibliography of Life is very exciting, and the author is to be highly commended both for clearly explaining that vision in this article, and also for his excellent work to make it a reality. This article itself is sure to encourage additional work in this area.

However, it would be most helpful for the general reader if the author could provide two additional summary tables of the information and conclusions presented in this article, to be included in the Discussion section, the first listing the advantages of Wikidata as a repository for the Bibliography of Life, and the second listing the present limitations and quirks of Wikidata as a repository for the Bibliography of Life, and the steps that would be required to remedy these drawbacks.

Reviewer 2 ·

Basic reporting

The article uses clear and professional English

Experimental design

NA

Validity of the findings

NA

Additional comments

Major comments
This manuscript provides an interesting idea regarding the usage of Wikidata as a global repository and structured database for taxonomic publications. Moreover, it provides a lot of information on the structure of the Wikidata entities, and their potential for this task.

However, I was a bit baffled by the overall goal, structure, and narrative of this contribution. While the author argues that Wikidata can be such a curated taxonomic database (e.g. in the abstract). It ends the MS by claiming that “Consequently a large number of taxonomic works and their authors already exist in Wikidata. As more and more taxonomic publications acquire DOIs, and as more working taxonomists acquire ORCID ids, the taxonomic literature component of Wikidata will automatically grow as content linked to these identifiers is routinely harvested by Wikidata bots. This leaves a large fraction of the taxonomic literature to be added by other means, but as discussed here there are numerous ways to do that. It is not unreasonable to expect that the bulk of the taxonomic literature will find its way into Wikidata in the next few years.”. So what are we to gain from this contribution? An acknowledgment of this fact? More understanding of the inner working of Wikidata? A call to use this source for this purpose? I think that I eventually got lost in this forest. Currently I finished reading this piece and had a hard time thinking what I have actually gained from it beyond the fact that Wikidata can be used as a taxonomic bibliography source – which you could probably state in much fewer words.

Linked to this critique I found your very limited discussion of the (many potential) drawbacks of Wikidata use for this purpose particularly lacking. I would greatly limit and reduce the current narrative of the MS, remove repetition in it and have a comprehensive section on drawbacks and limitations of Wikidata as a curated taxonomic bibliography source (and how these could potentially be addressed).

Perhaps you can structure both your main ideas, gaps to be filled, and potential limitations as a conceptual figure, that would help illuminate your statement(s). This together with much refocusing of the MS towards a specific agenda/target could greatly improve it.

Moreover, I think that claiming that your concept of Wikidata usage for this purpose is the “bibliography of life” a bit grandiose and I’d remove it from the title onwards. Having a central curated, structured, and interlinked, repository of taxonomic publications is a very important and impressive task, but hardly the “bibliography of life” this will be a bibliography of taxonomic works.

Lastly, the presentation of the author’s online tool in the MS does not seem to contribute to other potential messages found in it.

Minor comments
Line 65: Space missing after parenthesis

Figure 10 – I think there is a mistake in the node linking Wikispecies and zoobank, as the node between them contains many more shared authors than those listed in zoobank itself

·

Basic reporting

Clear, well-written and interesting. This paper meets all requirements in terms of basic reporting.

Experimental design

This paper does not involve traditional hypothesis testing approach but instead is reads more like an opinion piece highlighting and advocating the use of Wikidata in this context. Thus it does not have the usual Methods section. I think that's fine. The structure and design of the paper is appropriate for its context and thesis.

Validity of the findings

Findings and conclusions are relevant, interesting and well-supported.

Additional comments

Great work on this paper overall. I enjoyed reading it and found it interesting and important. I have three comments on places where I think it could be improved:

1. Inclusion of the ALEC website. This website is interesting and clearly related to the topic. However, its introduction felt out of place within the context of the paper. The thesis of the paper is that Wikidata can be used as a “centralized, curated literature database [for biological taxonomy].” While the ALEC site is a tool for exploring this, it is not critical supporting this thesis. Furthermore, as a user seeking to understand the ALEC tool I probably would not think to look to this paper for site documentation (though referenced, the site is not directly named in the abstract).

I think it would make more sense to move the ALEC description and documentation out of the paper and put it elsewhere. A supplement or something similar to the online documentation for R packages could be an example for this. The paragraphs beginning with lines 323 and 331 and Figs 3-6 could all be moved to ALEC site documentation rather than being part of this paper.

2. The community of Wikidata editors and contributors. As the author points out the Wikidata community of editors is a benefit (line 455) and also in some ways a challenge (line 469). It is important to include some additional discussion of who constitutes this community. From Fig 9 its obvious that relatively few contributors may contribute disproportionately to the content. What biases might emerge from this? Wikipedia editors have significant geographic and demographic skews and some of the literature on this could be referenced as an example. A few sentences on this topic should be added to the section on "A community of editors".

3. Lines 476-481. The point about taxa vs. names is an important one but here feels thrown in at the last moment and not fully explained. It would be useful to add of couple of sentences earlier on in the paper clarifying what this is and explaining its potential impact.

Other smaller comments:

1. Fig 10. I found this figure useful, but I think the information could be displayed in a cleaner and more easily interpretable way. A matrix similar to a correlation plot could be one approach: https://www.displayr.com/how-to-create-a-correlation-matrix-in-r/

2. Acronyms and technical terms. There are many mentioned in the paper. Are all of them necessary? It would be helpful to remove any acronyms that are not totally necessary to avoid confusion and help the reader. Of those that are included several need additional introduction beyond what is currently provided. For example: what are "quickstatements" (line 377), CONSTRUCT queries (line 332), what does ZOBODAT (line 31) stand for, etc.? Even SPARQL may not be known to all readers and should probably be briefly defined.

---

## Round 0.2 · Major Revisions

Dear Dr. Page,

The reviewers found your manuscript was strongly improved. However, a reviewer suggests further changes to improve the manuscript. In particular, he/she suggests to suggest discuss drawbacks/downsides of Wikidata and to analyses future prospects. So, I encourage you to improve the manuscript according to tips of this reviewer. Please, respond point-to-point to the comments of reviewers to speed up the process of revision.

Once again, thank you for submitting your manuscript to PeerJ and we look forward to receiving your revision.

Sincerely,
Gabriele Casazza

·

Basic reporting

The revised paper is now suitable for publication. I have a very few minor suggestions for improvement noted below in 4. Additional comments.

Experimental design

The increase in scope from 1000 to 10,000 WikiData bibliographic entities satisfies my previous reservations.

Validity of the findings

No comment.

Additional comments

While the paper is now publishable, the following minor improvements would improve accuracy and help readability:
1 (PDF Line 23): "Much of the primary data about the planet’s biodiversity is contained in the taxonomic literature, a corpus that dates to the eighteenth century." Change "dates to" to "dates from".
2 (PDF Line 237): "If both the citing article and the cited article have Crossref DOIs, then this citation link may ultimately find its way into COCI, the OpenCitations index of open Crossref DOI-to-DOI citations (Heibi, Peroni & Shotton, 2019)." Now that COCI is being regularly updated at bi-monthly intervals, please for accuracy change the text to read "If both the citing article and the cited article have Crossref DOIs and the citing article references are submitted to Crossref, then this citation link will appear in the next update of COCI, the OpenCitations index of open Crossref DOI-to-DOI citations (Heibi, Peroni & Shotton, 2019)."
3 (Legend for Table 1): "Table 1. Current progress towards mapping taxonomic names to the source literature. For each database the table gives the number of names in each taxonomic database that have an associated bibliographic reference, the number of those references that have at least one external identifier, and the number references with external identifiers that are in Wikidata." Change to read: ". . . the number AND PERCENTAGE of those references that have at least one external identifier, and the number AND PERCENTAGE **OF** references with external identifiers that are in Wikidata. [IPNI: International Plant Name Index]." (The acronym needs defining in the Table Legend as well as in the text.)
4 (Table 1 content): "ION" in Table 1 is not defined and does not appear in the body text. It should be removed from Table 1.
5 (Table 1 content): For consistency of presentation, "30" should be changed to "30.0".

Reviewer 2 ·

Basic reporting

-

Experimental design

-

Validity of the findings

-

Additional comments

The revised version of this manuscript is improved. It has more structure, and I can actually follow its narrative. I feel it is still on the long side – but this is for the editor to decide.

I think that this contribution could benefit from some forward thinking regarding its topic – where the author thinks greater effort in the future should be placed to produce the best outcomes, and how these should be brought about.

Another comment – carried on from my previous review has to do with drawbacks / downsides of Wikidata as a repository of taxonomic publications? Unfortunately, not referring to my initial comment on this issue does not fill me with confidence about your expertise on this topic. Without thinking about this for more than a few seconds I can think of: ease of access, user interface, reputation, various biases (geographic, taxonomic, cultural, journal based, publication type), as well as expanding on some of the issues currently addressed in penultimate paragraph regarding the community nature of Wiki and its lack of centralized curation. Highlighting such issues and others (potentially with data to support them) will only serve to make this contribution more comprehensive and useful. Wikidata is not a panacea, it would be unwise to treat it as such.

Regarding the title and stressing throughout the term “bibliography of life” – I still maintain that this is a poor choice. I understand the desire to keep it – but this manuscript deals with a centralized repository of taxonomic papers. I think it would make much more sense to treat it as such, and this would not take anything away from this contribution. As a testament to this you now add a paragraph in the intro to explain why this particular title was chosen. This is not a good sign that the title should have been chosen to begin with.

Line 273 - so is Wikispecies the bibliography of life or Wikidata?

---

## Round 0.3 · accepted · Accept

Dear Dr. Page,

I checked the manuscript and I am very pleased to inform you that your article "Wikidata and the bibliography of life" is accepted for publication in the PeerJ. Congratulations!

Thank you for submitting your work to PeerJ.

Sincerely,
Gabriele Casazza